# ATTENTION LOCALIZATION THROUGH SEPARATOR TOKENS: UNLOCKING LONG NUMERICAL SEQUENCE PROCESSING IN LLMs

## ABSTRACT

Despite possessing massive context windows, Large Language Models (LLMs) exhibit a sharp decline in performance when processing long numerical sequences, a critical failure for precision-sensitive applications. We identify the root cause as the models' inability to focus attention on a manageable sequence segment, leading to dispersed attention and inaccurate results. To address this, we introduce **Sep**arate **N**umerical **S**equences (SepNS), a training-free inference framework that guides LLMs by strategically inserting separators into numerical inputs. This simple modification encourages a "separate and focus" strategy, which we verify through attention analysis showing that separators induce localized focus on distinct segments. Extensive experiments on nine high-performance LLMs show SepNS substantially boosts accuracy, achieving average gains of **35.6%** across all evaluated datasets with less overhead. Our work demonstrates that simple, structured input formatting acts as a powerful attention-focusing mechanism, unlocking long numerical processing capabilities in LLMs without any retraining.

## 1 INTRODUCTION

Recent advances in Large Language Models (LLMs) have dramatically expanded their contextual capacity, with some models now supporting context windows of up to millions of tokens (Anthropic, 2025; Gemini-Team, 2025; Meta-AI, 2025; Team et al., 2025a;b), enabling processing of increasingly complex numerical data in domains such as weather forecasting (Gao et al., 2025) and stock analysis (Huang et al., 2021). However, a large context window does not guarantee that LLMs can analyze long sequences flawlessly (Hosseini et al., 2025). Empirical evidence suggests that LLMs experience substantial performance degradation when processing inputs that exceed 10-20% of their maximum context length (Kuratov et al., 2024; Liu et al., 2024b; Li et al., 2025c). Even in basic tasks like counting 1's in the numerical sequences, LLMs' performance declines as the length increases. As Figure 1(A) shows, accuracy drops by up to 70% as sequences grow from 2–32 to 256–512 elements in 6 fundamental tasks with the vanilla method. These fundamental deficiencies severely limit the deployment of LLMs in precision-critical scenarios, where numerical errors can propagate and lead to system failures, highlighting the urgent need to enhance LLMs' long sequence processing capabilities for reliable data-intensive applications.

Existing approaches to enhance long-context processing can be broadly categorized into three paradigms. First, attention mechanisms have successfully extended context windows and accelerated inference (Leviathan et al., 2025; Lai et al., 2025; Liu et al., 2024a), but they fundamentally fail to resolve precision issues inherent in numerical sequence processing. Second, while content processing strategies such as summarization (Hosseini et al., 2025; Liu et al., 2025; Li et al., 2025b) and reordering (Peysakhovich & Lerer, 2023; Chen et al., 2024b) have shown promise for textual content, they are inherently incompatible with numerical sequences where order and completeness are mathematically critical. Third, specialized tokenization approaches (Yang et al., 2025b) require extensive retraining, which incurs prohibitive computational costs and risks degrading the model's general-purpose capabilities. These limitations motivate our central research questions:

*What fundamental factors limit LLMs' ability to process long numerical sequences? How can we enhance this capability without additional training?*

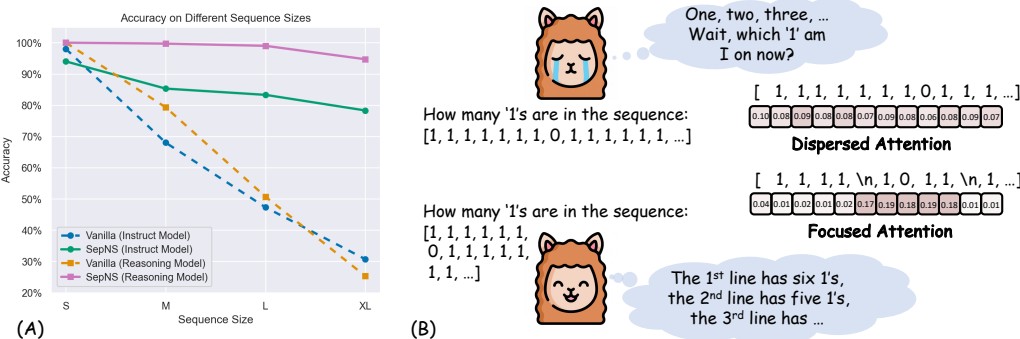

Figure 1: (A) Average accuracy across six synthetic tasks, performance drops sharply with increasing numerical sequence length (S: 2–32, M: 33–128, L: 129–256, XL: 257–512), while SepNS remains largely unaffected. (B) LLMs struggle with long numerical sequences due to dispersed attention, whereas SepNS uses separator tokens to maintain focused attention.

To answer these questions, we investigate the attention map for LLMs' sequence processing and analyze the underlying mechanisms behind their performance degradation. Our systematic analysis reveals a fundamental insight: LLMs' performance degradation on long numerical sequences stems from their limited capacity for focused attention on a separated sequence segment. As demonstrated in Figure 1(B), when confronted with long numerical sequence tasks, LLMs struggle to add natural breakpoints like humans do and process information in manageable chunks, leading to dispersed attention throughout the entire sequence. Based on this insight, we propose a simple yet effective approach to guide LLMs toward solving problems by introducing separators that partition long sequences into shorter, more manageable segments.

Building on these insights, we propose **Sep**arate **N**umerical **S**equences (SepNS), a training-free inference framework that enhances LLMs' long numerical sequence processing through strategic separator insertion. SepNS addresses the limitations through two key methodological contributions. First, we introduce systematic separator insertion, where separators are strategically placed within numerical sequences to create segmentation boundaries. This simple yet effective modification transforms intractable long sequences into multiple manageable segments that align with LLMs' reliable processing capacity, directly mitigating the precision degradation issue observed in Figure 1(A). Second, we conduct a comprehensive analysis of attention patterns to reveal the underlying mechanisms through both theoretical analysis and experimental validation. Our investigation shows that separators cause specific attention heads to focus predominantly on local sequence segments rather than dispersing attention across the entire sequence. This localized attention pattern enables more precise numerical processing while preserving global context by integrating information across segments.

We conduct comprehensive experiments across 9 high-performance LLMs, evaluating performance on six synthetic tasks and four real numerical sequence processing domains. Our results demonstrate that SepNS substantially outperforms both chain-of-thought (Wei et al., 2022) and one-shot (Yu et al., 2022) prompting strategies, achieving significant average accuracy gains of 35.6% across all evaluated datasets. Notably, these performance gains are achieved with reduced computational overhead: the method requires no additional training and actually reduces the inference burden compared to baseline approaches.

This work makes three key contributions to enhancing LLMs' capabilities for processing long numerical sequences. First, we systematically identify and characterize the fundamental bottleneck limiting LLMs' performance on long numerical sequences: dispersed attention to the entire numerical sequence. Second, we introduce SepNS, a training-free framework that mitigates this issue by strategically inserting separators into the input. Third, we provide comprehensive theoretical guarantees and empirical validation across diverse datasets, establishing the effectiveness, efficiency, and interpretability of our approach. Our findings demonstrate that simple input format changes can unlock substantial performance gains in LLMs' long numerical sequence processing.

## 2 RELATED WORK

**Progress and Challenges in Long-Text Processing.** Although LLMs now support millions of tokens (Anthropic, 2025; Gemini-Team, 2025; Team et al., 2025b), they still face significant challenges in long-context processing. Liu et al. (2024b) discover the "Lost in the Middle" phenomenon, where LLMs ignore key information in the middle sections of long documents. To address this, researchers propose solutions like MiniCache (Liu et al., 2024a) for efficient KV cache compression. And Hosseini et al. (2025) emphasizes that simply increasing context windows cannot guarantee perfect long sequence analysis.

**Deficiencies in Numerical Understanding.** Unlike semantic tasks, numerical tasks require understanding numbers as continuous quantities rather than discrete symbols, creating a fundamental mismatch with LLMs' token-based processing architecture. Research reveals that LLMs struggle with maintaining numerical precision in long sequences, often exhibiting digit transposition errors and magnitude confusion (Li et al., 2025a). Studies on mathematical reasoning demonstrate that LLMs struggle with multi-step numerical calculations (Wei et al., 2022). Tokenization approaches specifically designed for numerical data require extensive retraining, which is associated with prohibitive costs (Kudo & Richardson, 2018). Furthermore, research on numerical robustness reveals that models often fail to maintain precision in long numerical sequences (Hendrycks et al., 2021), highlighting the need for specialized approaches to enhance numerical understanding capabilities.

**Limitations of Tool Calling.** While tool calling capabilities of LLMs have rapidly developed through frameworks like ReAct (Yao et al., 2023) and Toolformer (Schick et al., 2023), they struggle with tasks requiring multi-source information integration. Studies show that even with access to external APIs and code execution environments (Gao et al., 2023), models face challenges in complex reasoning scenarios that demand synthesis of heterogeneous information sources (Press et al., 2022). Additionally, models exhibit reliability issues in long sequence processing, as reported by Welleck et al. (2019), who noted systematic failures in maintaining sequence fidelity. Advanced models also show unexpected character insertions during numerical sequence tasks, indicating persistent challenges in precise sequence tasks (Zhang et al., 2022).

## 3 METHOD

In this section, we begin by formally defining the problem of processing a long numerical sequence. Through preliminary experiments, we then demonstrate that LLMs exhibit significant limitations in accurately repeating such sequences. To address this critical issue, we propose **Sep**arate **N**umerical **S**equences (SepNS), a training-free and plug-and-play method designed to enhance LLMs' performance by strategically separating the input sequences. Finally, we provide a theoretical explanation for the effectiveness of our proposed method.

### 3.1 PROBLEM DEFINITION

We define a class of numerical sequence reasoning problems in which, given a numerical sequence $s = \{a_1, a_2, ..., a_n\}$ of length $|s| = n$, the task is to answer a natural language query $q$ based on this sequence. This class of problems is characterized by three core properties:

**Completeness Dependency.** Correct problem solving requires complete and accurate access to the entire numerical sequence. Any absence, modification, or omission of any element $a_i$ may lead to deviations or errors in the final answer.

**Natural Language Understanding Requirement.** Questions are posed in natural language, which requires accurate comprehension of user intent, including implicit conditional constraints, temporal scope limitations, and domain-specific semantic meanings.

**Composite Reasoning Complexity.** The tasks exhibit high complexity due to the need for integrating natural language understanding with numerical computation, involving conditional filtering, sequential pattern recognition, and multi-step logical reasoning.

For instance, given a stock trading data sequence, the question *"Excluding non-trading days, how many times did the open price of stock rise for three or more consecutive days?"* requires the



Figure 2: (A) and (C) visualize the attention scores given the input: "... How many 1's in [0 1 0 1 0 1 0 0 0] ...". (B) and (D) show the attention scores for the input with segmentation: "... How many 1's in [0 1 0 1 \n 0 1 0 0 \n 0] ...". (A) exhibits dispersed attention across the entire sequence. (B) demonstrates segment-focused attention. (C) highlights heightened attention at the beginning and end of the numerical sequence. (D) shows increased attention allocation to the separator token.

model to simultaneously understand the constraint condition of "excluding non-trading days", the sequential pattern of consecutive rises, and the counting requirement of three or more days.

## 3.2 THE REPETITION DILEMMA

Existing research (Dong et al., 2025; Pimentel et al., 2025; Junchi Yao, 2025) has shown that LLMs are prone to significant repetition errors. To quantify this limitation in the context of numerical sequences, we conduct preliminary experiments requiring LLMs to reproduce numerical sequences word-for-word. Our findings reveal systematic performance degradation as sequence length increases (see Appendix B for details), with a particularly steep decline observed beyond a critical threshold. Specifically, when sequences exceed 256 floating-point numbers, only 14% of attempts successfully repeat the sequence. This dramatic performance decline reveals the root cause of the long numerical sequence processing limitation, which stems from the LLM's limited capacity for understanding long numerical sequences, leading to frequent failures in preserving sequence integrity. This inherent deficiency ultimately results in the inability to construct accurate function calls or answer queries correctly, regardless of the availability of external tools.

## 3.3 ATTENTION LOCALIZATION THROUGH SEPARATOR TOKENS

To understand the underlying mechanisms behind these failures, we analyze attention patterns during the processing of long numerical sequences. As shown in Figure 2(A), we observe that LLMs tend to distribute attention across the entire sequence when processing a long numerical sequence. This behavior differs significantly from the "divide-and-conquer" strategy employed by humans when processing long sequences. Humans typically partition long sequences into several segments and process them sequentially with focused attention. Figure 2(C) reveals an interesting finding that while LLMs distribute attention across the entire sequence, attention weights tend to concentrate at the beginning and end of sequences. This finding aligns with research from Chen et al. (2024a), which demonstrates that LLMs exhibit attention concentration on certain special tokens (*e.g.,* start/end markers in sequences, punctuation marks in sentences, and other separators); furthermore, the semantic embedding vectors of these separators often encapsulate key information from their preceding segments. Based on these observations, we propose SepNS that guides LLMs to focus attention on local segments rather than the global sequence by artificially introducing specific separators into sequences.

Formally, given a numerical sequence $s$ of length $n$, SepNS transforms it into a structured format by periodically inserting separators. We define the transformation function as follows:

$$\text{SepNS}(s, k) = \{a_1, a_2, \ldots, a_k\} \oplus sep \oplus \{a_{k+1}, \ldots, a_{2k}\} \oplus sep \oplus \cdots \oplus \{a_{n-r+1}, \ldots, a_n\}, \quad (1)$$

where $k$ denotes the segment size, $sep$ represents the separator token (*e.g.,* "\n"), $\oplus$ denotes the concatenation operation, and $r$ is the remainder of $n$ divided by $k$. After introducing separators into sequences, we observe that transformer models exhibit a distinctive attention pattern: certain attention heads ignore contextual information before separators, and focus attention on the current separator and the numerical sequence following it, as illustrated in Figure 2(B, D).

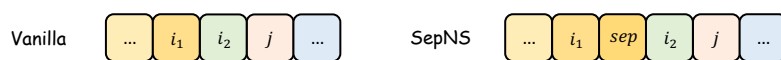

Figure 3: The separator token $sep$ in SepNS summarizes the content up to the current position. It exhibits a high score $\mathbf{Q}_{i_1} \cdot \mathbf{K}_{sep}^T$ for token $i_1$ (within its segment), but a low score $\mathbf{Q}_{i_2} \cdot \mathbf{K}_{sep}^T$ for token $i_2$ outside the summarized segment.

### 3.4 THEORETICAL EXPLANATION

We analyze how separator tokens mechanistically alter attention computation between tokens in different segments. For vanilla attention without separators, given sequence $s = \{a_1, a_2, \ldots, a_n\}$, the attention weight between positions $i$ and $j$ is computed using query vector $\mathbf{Q}_i$, key vector $\mathbf{K}_j$, and key dimension $d_k$:

$$A_{\text{vanilla}}[i,j] = \frac{\exp\left(\mathbf{Q}_i \cdot \mathbf{K}_j^T / \sqrt{d_k}\right)}{\sum_{l=1}^n \exp\left(\mathbf{Q}_i \cdot \mathbf{K}_l^T / \sqrt{d_k}\right)}. \tag{2}$$

With SepNS, the sequence becomes $s' = \{a_1, \ldots, a_k, sep, a_{k+1}, \ldots, a_{2k}, sep, \ldots\}$ with length $n'$. The attention weight between positions $i$ and $j$ is:

$$A_{\text{SepNS}}[i,j] = \frac{\exp\left(\mathbf{Q}_i \cdot \mathbf{K}_j^T / \sqrt{d_k}\right)}{\sum_{l=1}^{n'} \exp\left(\mathbf{Q}_i \cdot \mathbf{K}_l^T / \sqrt{d_k}\right)}. \tag{3}$$

Denote $S_{sep}$ as the set of all tokens before the current separator $sep$. As shown in Chen et al. (2024a), separator tokens $sep$ summarize tokens up to the current position, thus $sep$ exhibit high attention scores $\mathbf{Q}_{i_1} \cdot \mathbf{K}_{sep}^T$ for tokens $i_1 \in S_{sep}$ but low scores $\mathbf{Q}_{i_2} \cdot \mathbf{K}_{sep}^T$ for token $i_2 \notin S_{sep}$ (Figure 3). When computing attention from position $i_1 \in S_{sep}$ to position $j \notin S_{sep}$, separator tokens dramatically increase the denominator of $A_{\text{SepNS}}[i,j]$ through terms $\sum_{i_1 \in S_{sep}} \exp\left(\mathbf{Q}_{i_1} \cdot \mathbf{K}_{sep}^T / \sqrt{d_k}\right)$.

Thus for token $i_1 \in S_{sep}$ and $j \notin S_{sep}$, the cross-segment attention $A_{\text{SepNS}}[i_1,j]$ becomes significantly suppressed as:

$$\frac{A_{\text{SepNS}}[i_1,j]}{A_{\text{vanilla}}[i_1,j]} = \frac{\sum_{l=1}^n \exp\left(\mathbf{Q}_{i_1} \cdot \mathbf{K}_l^T / \sqrt{d_k}\right)}{\sum_{l=1}^{n'} \exp\left(\mathbf{Q}_{i_1} \cdot \mathbf{K}_l^T / \sqrt{d_k}\right)} \ll 1. \tag{4}$$

In contrast, for tokens $i_2 \notin S_{sep}$, $A_{\text{SepNS}}[i_2,j]/A_{\text{vanilla}}[i_2,j] \approx 1$ since $\exp\left(\mathbf{Q}_{i_2} \cdot \mathbf{K}_{sep}^T / \sqrt{d_k}\right)$ is small, maintaining high attention to tokens in the current segment.

This mathematical analysis reveals the underlying mechanism: separator tokens act as "attention sinks" that absorb attention weight otherwise dispersed to cross-segment positions. The high query-key similarity between tokens (*e.g.*, $i_1$) within the summarized segment and separator tokens (*e.g.*, $sep$) effectively "shields" these positions from attending to distant segments (*e.g.*, segment with $i_2, j$), thereby localizing attention within current segments and creating structured attention boundaries without explicit masking. See Appendix F for a detailed proof.

## 4 EXPERIMENTS

To evaluate our approach, we structure experiments around the following research questions (RQs), examining performance gains and robustness.

**RQ1 – Effectiveness.** Does our proposed method consistently enhance model performance across diverse tasks and model architectures, demonstrating its general applicability?

**RQ2 – Robustness.** Do any factors modulate the effectiveness of our method, and what actionable insights can we provide for optimal deployment in different scenarios?

In this section, we present a comprehensive experimental evaluation designed to systematically address these research questions. We begin by detailing our experimental setup, including the carefully curated datasets, evaluation metrics, selected LLMs, and baseline methods. We then present our experimental results and analyses across multiple dimensions to address the above research questions, demonstrating the superior performance of our method.

## 4.1 EXPERIMENTAL SETTINGS

### 4.1.1 DATASET

We design two datasets to conduct an in-depth investigation of LLMs' capabilities for processing long numerical sequences: a synthetic dataset $\mathcal{D}_{\text{syn}}$ and a real dataset $\mathcal{D}_{\text{real}}$.

For $\mathcal{D}_{\text{syn}}$, we construct sequences of varying lengths comprising both integer and floating-point numbers. We categorize these sequences into four length intervals: S (short) for sequences containing [2, 32] numbers, M (medium) for (32, 128] numbers, L (large) for (128, 256] numbers, and XL (extra-large) for (256, 512] numbers. We formulate six distinct task types: (1) *max-int*, which requires identifying the index of the maximum integer in an integer sequence; (2) *min-int*, which locates the index of the minimum integer; (3) *max-float* and (4) *min-float*, which perform analogous operations on floating-point sequences; (5) *indexing*, which determines the position of the last occurrence of 1 in a binary sequence; and (6) *counting*, which counts the total number of 1s in a binary sequence. Each task comprises 200 samples, with 50 samples distributed across each of the four length categories.

Building upon the prior work of Li et al. (2025a), we construct $\mathcal{D}_{\text{real}}$ to assess model performance on practical numerical reasoning tasks. This dataset comprises four distinct categories, each containing 200 samples. The categories include: (1) *number-string*, which involves counting numerals in alphanumeric sequences; (2) *number-list*, requiring logical reasoning over numerical sequences; and (3-4) *stock* and *weather*, both constructed from real-world datasets with human-generated questions. Notably, any failure to process a single value in these tasks inevitably results in an incorrect final answer, making them particularly challenging. See Appendix C for details and examples.

### 4.1.2 EVALUATION METRICS

To comprehensively evaluate our proposed method, we assess both performance and robustness using the following metrics:

**Accuracy (Acc).** This is our primary performance metric, measuring the percentage of correctly answered questions. Let $N_{\text{correct}}$ denote the number of responses that answer correctly out of $N$ total questions. Accuracy is computed as:

$$\text{Accuracy} = N_{\text{correct}}/N. \tag{5}$$

**Answer Rate (AR).** This metric captures the model's capability to generate valid responses across all test instances. We observe that models may occasionally fail to produce meaningful output, resulting in null responses. AR quantifies the proportion of questions for which the model generates a valid, non-null response. Let $N_{\text{valid}}$ denote the number of valid responses out of $N$ total questions:

$$\text{Answer Rate} = N_{\text{valid}}/N. \tag{6}$$

### 4.1.3 BASE MODELS

We conduct a comprehensive evaluation across 9 LLMs, representing diverse architectures, parameter scales, and training paradigms from both open-source and proprietary domains. **Open-source models:** Our selection includes the Qwen3 family (Yang et al., 2025a), spanning 0.6B to 30B parameters with both dense and Mixture-of-Experts architectures (Fedus et al., 2022; Zhou et al., 2023), available in instruct and reasoning modes, alongside the QwQ-32B model. We also evaluate the DeepSeek series (Guo et al., 2025), including the recent DeepSeek-R1 and DeepSeek-V3 variants, which are known for their strong reasoning capabilities. **Proprietary models:** We assess Claude-3.7-Sonnet from Anthropic (Anthropic, 2025). Additionally, we evaluate Google's Gemini-2.5-Pro (Gemini-Team, 2025), which showcases multimodal understanding capabilities, and two variants from OpenAI's GPT-4 series (Achiam et al., 2023): GPT-4.1 and GPT-4o. See Section 7 for a detailed version of the models.

### 4.1.4 BASELINES

To establish comprehensive benchmarks for evaluating our proposed method, we employ two baseline approaches except the vanilla method. **Chain-of-Thought:** In the chain-of-thought setting (Wei

Table 1: The average answer rate and accuracy of each task among the four methods. "Incr." indicates the percentage improvement of SepNS over Vanilla. Green and red indicate improvement and degradation in performance metrics, respectively.

| Task | Answer Rate (↑) | | | | Accuracy (↑) | | | |
|---|---|---|---|---|---|---|---|---|
| | Vanilla | CoT | One-shot | SepNS (Incr.) | Vanilla | CoT | One-shot | SepNS (Incr.) |
| counting | 100.0% | 100.0% | 100.0% | 100.0% ( +0.0%) | 42.7% | 41.2% | 37.9% | 76.7% ( +79.6%) |
| indexing | 100.0% | 100.0% | 100.0% | 100.0% ( +0.0%) | 38.8% | 33.4% | 34.9% | 86.6% (+123.0%) |
| max-float | 100.0% | 100.0% | 100.0% | 100.0% ( +0.0%) | 63.9% | 60.5% | 63.3% | 81.0% ( +26.8%) |
| max-int | 100.0% | 100.0% | 100.0% | 100.0% ( +0.0%) | 75.3% | 68.8% | 71.0% | 92.4% ( +22.7%) |
| min-float | 100.0% | 100.0% | 100.0% | 100.0% ( +0.0%) | 63.1% | 60.2% | 60.4% | 79.3% ( +25.7%) |
| min-int | 100.0% | 100.0% | 100.0% | 100.0% ( +0.0%) | 74.3% | 68.1% | 66.5% | 93.0% ( +25.1%) |
| number-string | 96.6% | 96.3% | 96.3% | 99.0% ( +2.5%) | 81.6% | 81.0% | 81.7% | 81.7% ( +0.1%) |
| number-list | 78.2% | 77.6% | 73.4% | 79.1% ( +1.1%) | 36.3% | 35.4% | 32.6% | 36.7% ( +0.9%) |
| stock | 64.1% | 63.1% | 63.3% | 73.6% (+14.7%) | 13.2% | 13.7% | 13.4% | 27.4% (+107.6%) |
| weather | 66.0% | 65.6% | 65.9% | 76.8% (+16.3%) | 26.7% | 27.2% | 27.8% | 44.6% ( +67.1%) |
| Average | 90.5% | 90.3% | 89.9% | **92.8%** ( +2.6%) | 51.6% | 49.0% | 48.9% | **69.9%** ( +35.6%) |

et al., 2023), the model is guided by prompt instructions to complete the task step by step. **One-shot Learning:** For the one-shot baseline (Yu et al., 2022), we provide the model with a single demonstration example that illustrates the desired input-output mapping for the target task.

## 4.2 EXPERIMENTAL RESULTS

In this section, we present a comprehensive evaluation designed to systematically address our research questions. Each research question is analyzed and substantiated through multiple complementary perspectives, providing thorough empirical evidence for our claims.

### 4.2.1 EFFECTIVENESS (RQ1)

**Across tasks.** We evaluate our method against baselines on six synthetic and four real tasks, reporting Accuracy and Answer Rate (additional metrics (*e.g.,* "Response Length") in Appendix D). As shown in Table 1, our analysis reveals critical deficiencies in standard LLMs when processing long numerical sequences. Notably, popular enhancement methods, such as Chain-of-Thought (CoT) and One-shot prompting, fail to address this fundamental problem. In fact, they prove detrimental, with average accuracies dropping from 51.6% (Vanilla) to 49.0% (CoT) and 48.9% (One-shot). This strongly suggests that LLM's failures stem not from an insufficient reasoning ability but from a fundamental inability to properly parse and manage numerical sequences.

In contrast, our SepNS framework demonstrates remarkable efficacy, substantially elevating performance across all tasks. By strategically structuring input, SepNS boosts average accuracy to 69.9%, a significant 35.6% relative improvement over the vanilla baseline. This improvement extends beyond synthetic data—on real tasks, complex numerical tasks, SepNS proves equally effective. Accuracy on stock and weather datasets increases dramatically by +107.6% and +67.1%, respectively. SepNS also enhances reliability, increasing average answer rate to 92.8% (+2.6%), ensuring models provide both more accurate and consistent responses. These results confirm that SepNS effectively rectifies core weaknesses in LLM sequence understanding, enabling significant improvements in numerical processing without requiring model modifications.

**Across models.** We evaluate 9 diverse high-performance LLMs and summarize model-wise performance gains over vanilla baselines. Results are presented in model-wise tables to validate the general applicability of our approach across different model architectures and scales.

The results in Table 2 demonstrate that SepNS provides consistent performance enhancements across all tested architectures, achieving a remarkable +35.6% average accuracy boost over the vanilla baseline. The framework yields significant gains irrespective of model size or origin. Open-source models like QwQ-32B and Qwen3-8B exhibit significant improvements of +69.0% and +53.0%, respectively, suggesting SepNS effectively unlocks numerical processing capabilities. This trend extends to advanced proprietary models—Claude-3.7-Sonnet (+38.0%), Gemini-2.5-Pro (+34.3%), and GPT-4.1 (+35.6%), all of which derive significant benefits. These indicate that difficulty handling long numerical sequences is a fundamental limitation inherent in current LLM architectures

Table 2: The average answer rate and accuracy of each model among the four methods. "Incr." indicates the percentage improvement of SepNS over Vanilla. Green and red indicate improvement and degradation in performance metrics, respectively.

| Model | Answer Rate (↑) | | | | Accuracy (↑) | | | |
|---|---|---|---|---|---|---|---|---|
| | Vanilla | CoT | One-shot | SepNS (Incr.) | Vanilla | CoT | One-shot | SepNS (Incr.) |
| Qwen3-8B | 79.4% | 79.5% | 75.2% | 87.3% ( +9.9%) | 45.5% | 40.3% | 38.5% | 69.6% (+53.0%) |
| Qwen3-30B-A3B | 80.6% | 80.9% | 81.0% | 90.2% (+11.9%) | 54.3% | 52.7% | 52.3% | 80.7% (+48.6%) |
| QwQ-32B | 72.7% | 72.2% | 72.7% | 75.6% ( +4.0%) | 34.2% | 28.9% | 28.5% | 57.8% (+69.0%) |
| DeepSeek-V3 | 100.0% | 100.0% | 100.0% | 100.0% ( +0.0%) | 45.7% | 44.5% | 45.2% | 50.9% (+11.4%) |
| DeepSeek-R1 | 99.9% | 99.9% | 99.8% | 99.9% ( +0.0%) | 61.1% | 56.6% | 57.9% | 70.5% (+15.4%) |
| Claude-3.7-Sonnet | 99.8% | 99.9% | 100.0% | 99.9% ( +0.1%) | 57.3% | 57.6% | 54.2% | 79.1% (+38.0%) |
| Gemini-2.5-Pro | 83.4% | 81.8% | 82.0% | 84.6% ( +1.4%) | 58.9% | 48.0% | 56.9% | 79.1% (+34.3%) |
| GPT-4.1 | 99.6% | 99.6% | 99.8% | 99.5% ( -0.1%) | 61.0% | 60.0% | 60.0% | 82.7% (+35.6%) |
| GPT-4o | 99.0% | 98.5% | 98.6% | 98.6% ( -0.4%) | 46.4% | 51.5% | 44.5% | 59.1% (+27.4%) |
| Average | 90.5% | 90.3% | 89.9% | **92.8%** ( +2.6%) | 51.6% | 48.9% | 48.7% | **68.9%** (+35.6%) |

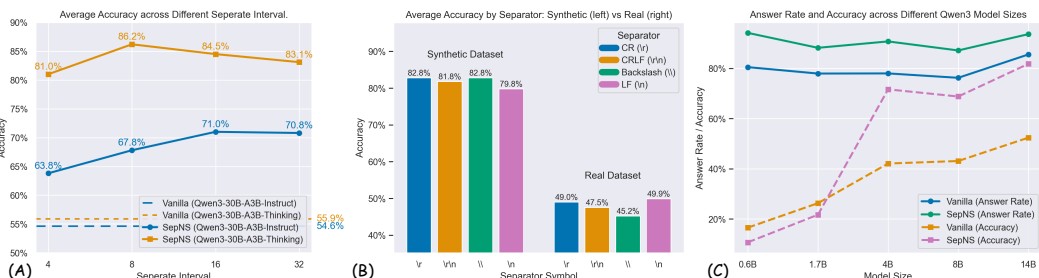

Figure 4: (A): Average Accuracy across different separate intervals. (B): Average Accuracy across different separator symbols. (C): Average Answer Rate and Accuracy across different Qwen3 model sizes, ranging from 0.6B to 14B.

that cannot be overcome merely by increasing model scale or training data. While some models like DeepSeek show more modest gains, improvements remain consistently positive. Furthermore, SepNS enhances model reliability. For models with lower initial answer rates, such as Qwen3-8B and Qwen3-30B-A3B, our method provides notable increases of +9.9% and +11.9%, respectively, improving dependability for practical applications. The widespread performance uplift across diverse LLMs strongly supports our thesis: SepNS effectively guides LLMs to better comprehend and process structured numerical data. Tables 6 and 7 in Appendix E.1 further demonstrate that SepNS reduces the average number of response tokens and decreases computing resource requirements.

## 4.3 ROBUSTNESS (RQ2)

**Separate interval.** We investigate the impact of separator intervals and report how performance varies, revealing a clear pattern that highlights the fundamental trade-off between providing structural guidance and maintaining contextual integrity.

Figure 4(A) shows SepNS on both Qwen3-30B-A3B models outperforms vanilla baselines (54.6% and 55.9%), confirming fundamental benefits. However, performance is highly interval-sensitive. The reasoning model peaks at 86.2% (interval 8), while the instruction model reaches 71.0% (interval 16), suggesting optimal moderate sparsity. Too-short intervals (4) overly fragment sequences, preventing local relationship capture, while too-long intervals (32) weaken structural cues. This demonstrates that SepNS effectiveness requires strategic separator insertion. The "sweet spot" provides practical guidelines for default interval selection (8 or 16), maximizing accuracy while minimizing token overhead. Table 8 in Appendix E.2 gives detail results.

**Separator symbol.** We investigate separator symbol impact on performance using Qwen3-30B-A3B-Instruct-2507 across 10 tasks with four separator types: Carriage Return (CR, \r), Carriage Return Line Feed (CRLF, \r\n), Backslash (\\), and Line Feed (LF, \n). Tasks were categorized into basic numerical processing ($\mathcal{D}_{syn}$) and complex applications ($\mathcal{D}_{real}$).

Table 3: Accuracy comparison between instruction (Qwen3-30B-A3B-Instruct) and reasoning (Qwen3-30B-A3B-Thinking) models across different separate intervals (4/8/16/32). **Bold** and underlined values denote the highest and second-highest scores, respectively.

| Size | Accuracy on Instruct Model (↑) | | | | | Accuracy on Reasoning Model (↑) | | | | |
|------|---------|------|------|------|------|---------|------|------|------|------|
|      | vanilla | 4 | 8 | 16 | 32 | vanilla | 4 | 8 | 16 | 32 |
| S  | **98.0%** | 88.7% | 91.0% | 94.0% | 95.0% | **100.0%** | **100.0%** | 99.7% | **100.0%** | 99.3% |
| M  | 68.0% | 81.7% | 81.0% | **85.3%** | 81.0% | 79.3% | 99.6% | **100.0%** | 99.7% | 99.7% |
| L  | 47.3% | 70.3% | 78.7% | **83.3%** | 79.7% | 50.7% | 98.0% | **99.3%** | 99.0% | 97.0% |
| XL | 30.7% | 65.3% | 68.7% | 78.3% | **80.7%** | 25.3% | 81.2% | 92.0% | **94.7%** | 91.7% |
| Avg. | 61.0% | 76.5% | 79.8% | **85.2%** | 84.1% | 63.8% | 94.7% | 97.7% | **98.3%** | 96.9% |

Figure 4(B) reveals systematic performance differentiation across separator types varying with task complexity. For basic tasks $\mathcal{D}_{syn}$, all separators showed modest accuracy variations (79.8%–82.8%), with unconventional separators CR and Backslash achieving optimal accuracy (82.8% each), potentially due to novelty requiring enhanced attention. However, inversion emerged in complex scenarios: while CR and Backslash excelled in basic tasks, LF demonstrated superior performance (49.9% vs. Backslash's 45.2%) in complex tasks. This reversal suggests fundamental processing strategy shifts across complexity levels. These patterns evidence sophisticated cognitive resource allocation (Sweller, 1988; Sweller et al., 2011) in LLMs. Under low cognitive load, models allocate additional resources to separator adaptation, where novel separators benefit from enhanced attention. Under high cognitive load, models prioritize core semantic processing, favoring minimal-overhead separators. LF's superior complex performance reflects the prevalence of training data and processing efficiency, enabling more resources for task-specific reasoning over format adaptation.

**Model size.** We evaluated Qwen3 models at various parameter scales (0.6B, 1.7B, 4B, 8B, and 14B). The results, shown in Figure 4(C), demonstrate that the effectiveness of SepNS is scale-dependent. For smaller models (0.6B), SepNS underperforms the vanilla baseline, suggesting a minimum capacity requirement for effective separator interpretation. A critical inflection point occurs at the 4B parameter level, where SepNS begins to yield performance gains. Beyond this scale, the performance gap widens substantially, reaching over 80% accuracy on the 14B models.

**Reasoning vs. Instruction models.** We contrast reasoning variants against the instruction model, reporting paired differences to assess whether reasoning ability affects separator sensitivity. Table 3 reveals powerful effects when applying SepNS to stronger reasoning models.

With the vanilla method, both model types suffer severe degradation on long sequences. On extra-long sequences, vanilla accuracy drops to 30.7% (instruction) and 25.3% (reasoning), highlighting shared weaknesses. However, SepNS reveals significant gaps: instruction models reach 85.2% peak accuracy (interval 16), while reasoning models achieve a near-perfect 98.3%. On large sequences, reasoning accuracy jumps from 50.7% to 99.3%, versus instruction's modest 47.3% to 83.3% improvement. Reasoning models exhibit greater interval robustness, maintaining near-optimal performance across intervals 8-32, while instruction models are more hyperparameter-sensitive. This indicates SepNS provides structural decomposition that reasoning models uniquely exploit, converting challenging problems into manageable sub-tasks for state-of-the-art accuracy.

## 5 CONCLUSION

In this work, we identify and address a fundamental limitation of LLMs: their inability to maintain focused attention when processing long numerical sequences, resulting in severe performance degradation in precision-critical applications. We introduced SepNS, a training-free framework that strategically inserts separators to partition long sequences into manageable segments. Through comprehensive evaluation across 9 high-performance models on 10 tasks, SepNS achieves a substantial 35.6% average accuracy improvement without computational overhead or retraining. Our analysis reveals that separators induce localized attention patterns, transforming dispersed attention into focused segment processing while preserving global context. This demonstrates that simple input formatting serves as a powerful attention-focusing mechanism, unlocking significant numerical processing capabilities and providing practical solutions for precision-critical applications.

## 6 ETHICS STATEMENT

Our research adheres to the ICLR Code of Ethics and raises no ethical concerns. The proposed SepNS framework is a training-free inference technique that modifies input formatting without altering model parameters or requiring additional data collection. Our experiments utilize publicly available models and synthetic datasets, with no involvement of human subjects, collection of personal data, or privacy risks. The method enhances model accuracy in numerical processing tasks without introducing harmful capabilities or creating potential for misuse. All experimental evaluations were conducted using established benchmarks and standard evaluation protocols. The research makes a positive contribution to the field by addressing fundamental limitations in LLM numerical processing capabilities, with potential benefits for applications that require precise numerical computation.

## 7 REPRODUCIBILITY STATEMENT

To facilitate reproducibility of our results, we provide comprehensive documentation and resources across multiple components of this work.

**Code and Data.** Our proposed method is thoroughly detailed in Section 3, including algorithmic descriptions and implementation specifics. Complete source code and datasets are available through the anonymous repository at https://anonymous.4open.science/r/SepNS, with the accompanying README file providing step-by-step instructions for execution and reproduction of experiments.

**Theorem.** Rigorous theoretical foundations are established in Appendix F, which contains detailed mathematical proofs and derivations supporting our theoretical claims. These materials collectively provide researchers with the necessary resources to validate and build upon our contributions.

**Selected Models.** We evaluate diverse high-performance models across different experimental settings. For the **main evaluation (RQ1)**, we use 9 high-performance LLMs:

1. Qwen3-8B: https://huggingface.co/Qwen/Qwen3-8B

2. Qwen3-30B-A3B: https://huggingface.co/Qwen/Qwen3-30B-A3B

3. QwQ-32B: https://huggingface.co/Qwen/QwQ-32B

4. DeepSeek-V3: https://huggingface.co/deepseek-ai/DeepSeek-V3-0324

5. DeepSeek-R1: https://huggingface.co/deepseek-ai/DeepSeek-R1-0528

6. Claude-3.7-Sonnet: https://openrouter.ai/anthropic/claude-3.7-sonnet

7. Gemini-2.5-Pro: https://openrouter.ai/google/gemini-2.5-pro

8. GPT-4.1: https://openrouter.ai/openai/gpt-4.1

9. GPT-4o: https://openrouter.ai/openai/gpt-4o-2024-08-06

For robustness evaluation (RQ2), **separator interval analysis**, we compare instruction and reasoning variants:

1. Qwen3-30B-A3B-Instruct: https://huggingface.co/Qwen/Qwen3-30B-A3B-Instruct-2507

2. Qwen3-30B-A3B-Thinking: https://huggingface.co/Qwen/Qwen3-30B-A3B-Thinking-2507

For **separator symbol analysis**, we use Qwen3-30B-A3B-Instruct (https://huggingface.co/Qwen/Qwen3-30B-A3B-Instruct-2507).

For **model size analysis**, we evaluate across different parameter scales:

1. Qwen3-0.6B: https://huggingface.co/Qwen/Qwen3-0.6B

2. Qwen3-1.7B: https://huggingface.co/Qwen/Qwen3-1.7B

3. Qwen3-4B: https://huggingface.co/Qwen/Qwen3-4B

4. Qwen3-8B: https://huggingface.co/Qwen/Qwen3-8B

5. Qwen3-14B: https://huggingface.co/Qwen/Qwen3-14B

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

Table 4: Performance of Qwen3-30B-A3B-Instruct-2507 and Qwen3-30B-A3B-Thinking-2507 on the strict numerical sequence repetition task. The table shows the number of correct reproductions and accuracy for each sequence length range.

| Size | Total | Qwen3-30B-A3B-Instruct Model | | Qwen3-30B-A3B-Thinking-2507 | |
|------|-------|-----------|----------|-----------|----------|
| | | # Correct | Accuracy | # Correct | Accuracy |
| S: 2-32 | 50 | 50 | 100.00% | 50 | 100.00% |
| M: 33-128 | 50 | 50 | 100.00% | 36 | 72.00% |
| L: 129-256 | 50 | 50 | 100.00% | 16 | 32.00% |
| XL: 257-512 | 50 | 7 | 14.00% | 0 | 0.00% |
| XXL: 513-1024 | 50 | 0 | 0.00% | 0 | 0.00% |
| **Overall** | 250 | 157 | 62.80% | 102 | 40.80% |

## A    THE USE OF LARGE LANGUAGE MODELS

We employed LLMs for bug detection in code. Additionally, LLMs were utilized to refine and polish the manuscript content based on specific requirements. All LLM-generated content, including code and textual revisions, underwent thorough review and validation by the authors to ensure accuracy, quality, and alignment with our research objectives.

## B    THE EXPERIMENT OF REPETITION DILEMMA

To evaluate the capability of LLMs on repetition, we conducted a preliminary experiment on "strict numerical sequence repetition." The task requires a model to reproduce a given numerical sequence exactly, without any additions, omissions, or alterations. We designed a testing framework with progressively increasing difficulty by dividing sequence lengths into five ranges: 2–32, 33–128, 129–256, 257–512, and 513–1024. For each range, 50 unique samples were randomly generated. Each sequence consisted of numbers with three decimal places, drawn uniformly from the interval [-10, 10]. Models were prompted to return the output in a strict JSON array format (*e.g.,* [1.234, -5.678, 9.012]), prohibiting any extraneous characters, spaces, or line breaks. A response was judged as correct only if it was an exact string match to the ground truth sequence.

The experiment was performed on two variants of the Qwen3 model: Qwen3-30B-A3B-Instruct-2507 and Qwen3-30B-A3B-Thinking-2507. To ensure deterministic and stable outputs, the decoding temperature was set to 0. Any deviation in format or content from the expected output was classified as an error.

The results, presented in Table 4, reveal a strong "repetition dilemma" in both models. The Qwen3-30B-A3B-Instruct-2507 variant performed flawlessly on sequences up to 256 numbers, achieving 100% accuracy. However, its performance collapsed to just 14% accuracy in the 257–512 range (XL) and failed completely on the longest sequences (XXL). Counter-intuitively, the Qwen3-30B-A3B-Thinking-2507 variant, despite its designation, demonstrated inferior overall performance (40.80% vs. 62.80%). Its accuracy began to degrade significantly on medium-length sequences (M and L), falling far short of its instruct-tuned counterpart.

These findings highlight significant architectural or attentional limitations in current LLMs for tasks demanding precise, long-sequence replication. Such failures may stem from compounding errors in the attention mechanism, effective context window constraints, or biases in the training data. The inferior performance of the "Thinking" variant is particularly noteworthy. It suggests that for rote, mechanical tasks that do not require reasoning, the cognitive overhead or architectural modifications intended to facilitate complex thought may act as a source of noise, thereby degrading performance on simple memorization and reproduction.

## C    DATASET DETAILS

This appendix provides detailed descriptions of the datasets used to evaluate LLMs' long numerical sequence processing capabilities.

## C.1 SYNTHETIC DATASET

The synthetic dataset $D_{\text{syn}}$ consists of 1,200 samples across six task types, each containing 200 samples. The sequences are categorized into four length intervals:

- **S (Short)**: [2, 32] numbers (50 samples per task)
- **M (Medium)**: (32, 128] numbers (50 samples per task)
- **L (Large)**: (128, 256] numbers (50 samples per task)
- **XL (Extra-large)**: (256, 512] numbers (50 samples per task)

### C.1.1 TASK TYPES

**max-int**: Identify the index (0-based) of the maximum integer in an integer sequence. Example:

```
1  {
2    "task_type": "max_int",
3    "answer": "7",
4    "ts": [3, 2, 2, 0, 3, 0, 2, 5, 0, 0, 1, 0, 1, 0, 1, 2, 0, 1, 4]
5  }
```

The maximum value is 5 at index 7.

**min-int**: Identify the index (0-based) of the minimum integer in an integer sequence. Example:

```
1  {
2    "task_type": "min_int",
3    "answer": "19",
4    "ts": [6, 9, 7, 6, 7, 7, 6, 6, 6, 7, 9, 8, 7, 6, 6, 8, 8, 8, 9, 2, 9, 9, 6, 9, 6, 8
         , 6, 9, 6]
5  }
```

The minimum value is 2 at index 19.

**max-float**: Identify the index (0-based) of the maximum floating-point number in a sequence. Similar to max-int but with floating-point numbers.

**min-float**: Identify the index (0-based) of the minimum floating-point number in a sequence. Similar to min-int but with floating-point numbers.

**indexing**: Determine the position of the last occurrence of 1 in a binary sequence. Example:

```
1  {
2    "task_type": "indexing",
3    "answer": "8",
4    "ts": [1, 0, 0, 0, 1, 0, 0, 1, 1, 0, 0, 0, 0, 0, 0, 0, 0, 0, 0, 0, 0]
5  }
```

The last occurrence of 1 is at index 8.

**counting**: Count the total number of 1s in a binary sequence. Example:

```
1  {
2    "task_type": "counting",
3    "answer": "4",
4    "ts": [1, 0, 0, 0, 1, 0, 0, 1, 1, 0, 0, 0, 0, 0, 0, 0, 0, 0, 0, 0, 0]
5  }
```

There are 4 occurrences of 1 in the sequence.

## C.2    REAL DATASET

The real dataset $D_{\text{real}}$ consists of 800 samples across four categories, each containing 200 samples. These tasks are based on practical numerical reasoning scenarios.

### C.2.1    TASK CATEGORIES

**number-string**: Count numerals in alphanumeric sequences. Example:

```
1  {
2    "question": "How many numbers are there in the string? Note that a sequence like
         'a243b' counts as a single number.",
3    "struct_data": "effV2xM8hF5vcNgl8xrTCmbD6sEM38tiK4Nn2vem14f698o7Lo",
4    "answer": 11
5  }
```

This task requires parsing mixed alphanumeric strings to identify and count distinct numerical sequences.

**number-list**: Perform logical reasoning over numerical sequences with multiple-choice questions. Example:

```
1  {
2    "question": "Which index holds the greatest number in the list between the indices
         20 and 80? Options: A: 40, B: 75, C: 53, D: 58, E: 48, F: 44, G: 60, H: 31",
3    "struct_data": [1372.31, -3479.74, 1046, "..."],
4    "answer": "H"
5  }
```

These tasks involve complex reasoning operations such as finding extrema within specific ranges, identifying patterns, or performing conditional operations.

**stock**: Answer questions about financial time series data. Example:

```
{
  "question": "How many days had a volume over 15,000 between 2024-10-15 and
       2024-10-25? Options: A: 3, B: 5, C: 7, D: 9",
  "struct_data": [
    {"date": "2024-10-15", "close_price": 52.56, "volume": 24421, "...": "..."},
    {"date": "2024-10-16", "close_price": 52.80, "volume": 19962, "...": "..."},
    {"date": "2024-10-17", "close_price": 53.11, "volume": 19210, "...": "..."},
    {"date": "2024-10-18", "close_price": 55.11, "volume": 25238, "...": "..."},
    "..."
  ],
  "answer": "C"
}
```

Listing 1: Stock example (truncated)

This task involves analyzing real-world financial time series data with questions about trading volumes, price movements, and temporal patterns.

**weather**: Answer questions about meteorological time series data. Example:

```
{
  "question": "On which date was the temperature lastly above 5 degrees between
       2024-11-10 and 2024-11-20? Options: A: 2024-11-11, B: 2024-11-14, C:
       2024-11-12, D: 2024-11-13",
  "struct_data": [
    {"date": "2024-11-10", "temperature_2m": 5.99, "precipitation": 0.0, "...":
        "..."},
    {"date": "2024-11-11", "temperature_2m": 5.51, "precipitation": 0.0, "...":
        "..."},
    {"date": "2024-11-12", "temperature_2m": 5.10, "precipitation": 0.0, "...":
        "..."},
```

```
    {"date": "2024-11-13", "temperature_2m": 4.92, "precipitation": 0.0, "...":
        "..."},
    {"date": "2024-11-14", "temperature_2m": 5.45, "precipitation": 0.0, "...":
        "..."},
    "..."
],
"answer": "B"
}
```

Listing 2: Weather example (truncated)

This task involves analyzing real-world meteorological time series data with questions about temperature patterns, precipitation, and temporal trends.

### C.3 DATASET STATISTICS

Table 5 shows the dataset statistics.

Table 5: Dataset overview and statistics.

| Dataset | Task Types | Total Samples | Length Categories | Sample Distribution |
|---------|-----------|---------------|-------------------|---------------------|
| $D_{syn}$ | 6 | 1,200 | 4 (S/M/L/XL) | 50 per category per task |
| $D_{real}$ | 4 | 800 | Variable | 200 per task |

## D ADDITIONAL METRICS

SepNS adds extra separators, which results in a longer input length. Since the number of input and output tokens is directly related to the computational cost, we use metrics Input Length (IL), Response Length (RL) and Total Length (TL) defined as follows to quantify the mdoel performance:

**Input Length (IL)** is the average number of tokens in the input prompts. Let $L_{\text{prompt}_i}$ be the token count for prompt $i$:

$$\text{IL} = \frac{1}{N} \sum_{i=1}^{N} L_{\text{input}_i}. \tag{7}$$

**Response Length (RL)** is the average number of tokens in the generated responses. This serves as a proxy for inference cost. Let $L_{\text{response}_i}$ be the token count for the response to prompt $i$:

$$\text{RL} = \frac{1}{N} \sum_{i=1}^{N} L_{\text{response}_i}. \tag{8}$$

**Total Length (TL)** is the average number of tokens in the inputs and responses. This serves as a proxy for computing cost:

$$\text{TL} = \text{IL} + \text{RL}. \tag{9}$$

## E DETAILED EXPERIMENTAL RESULTS

### E.1 MAIN RESULTS WITH LENGTH

This experiment evaluated SepNS effectiveness across different task types. Results in Table 6 reveal that SepNS demonstrates consistent improvements in accuracy across most tasks, with particularly strong gains for information localization tasks like indexing (+123.0%) and counting (+79.6%), as well as real-world applications such as stock (+107.6%) and weather (+67.1%) analysis. The method also achieves solid improvements on numerical comparison tasks, with max-int (+22.7%), min-int (+25.1%), max-float (+26.8%), and min-float (+25.7%) all showing substantial accuracy gains. Notably, SepNS maintains perfect Answer Rates (100.0%) across all synthetic tasks while improving response efficiency, as evidenced by reduced response lengths for most tasks. The only exceptions are number-list tasks, which show minimal accuracy improvement (+0.9%) despite slight Answer

Table 6: Average performance comparison of different tasks (Vanilla vs SepNS). Green and red indicate improvement and degradation in performance metrics, respectively.

| Task | Answer Rate (↑) | | | Accuracy (↑) | | | Response Len (↓) | | | Total Len (↓) | | |
|---|---|---|---|---|---|---|---|---|---|---|---|---|
| | Vanilla | SepNS | Incr. | Vanilla | SepNS | Incr. | Vanilla | SepNS | Incr. | Vanilla | SepNS | Incr. |
| counting | 100.0% | 100.0% | +0.0% | 42.7% | 76.7% | +79.6% | 3004 | 2106 | -29.9% | 3594 | 2740 | -23.8% |
| indexing | 100.0% | 100.0% | +0.0% | 38.8% | 86.6% | +123.0% | 4035 | 2304 | -42.9% | 4573 | 2909 | -36.4% |
| max-float | 100.0% | 100.0% | +0.0% | 63.9% | 81.0% | +26.8% | 3459 | 3603 | +4.2% | 4544 | 4843 | +6.6% |
| max-int | 100.0% | 100.0% | +0.0% | 75.3% | 92.4% | +22.7% | 2242 | 1744 | -22.2% | 2741 | 2334 | -14.9% |
| min-float | 100.0% | 100.0% | +0.0% | 63.1% | 79.3% | +25.7% | 3676 | 3722 | +1.2% | 4764 | 4945 | +3.8% |
| min-int | 100.0% | 100.0% | +0.0% | 74.3% | 93.0% | +25.1% | 2338 | 1829 | -21.8% | 2843 | 2425 | -14.7% |
| number-string | 96.6% | 99.0% | +2.5% | 81.6% | 81.7% | +0.1% | 1855 | 1569 | -15.4% | 2059 | 1803 | -12.4% |
| number-list | 78.2% | 79.1% | +1.1% | 36.3% | 36.7% | +0.9% | 2881 | 4205 | +46.0% | 5643 | 7526 | +33.4% |
| stock | 64.1% | 73.6% | +14.7% | 13.2% | 27.4% | +107.6% | 5285 | 3721 | -29.6% | 13328 | 13173 | -1.2% |
| weather | 66.0% | 76.8% | +16.3% | 26.7% | 44.6% | +67.1% | 4972 | 3428 | -31.1% | 11901 | 11642 | -2.2% |
| Average | 90.5% | 92.8% | +2.6% | 51.6% | 69.9% | +35.6% | 3375 | 2823 | -16.3% | 5599 | 5434 | -2.9% |

Table 7: Average performance comparison of different models (Vanilla vs SepNS). Green and red indicate improvement and degradation in performance metrics, respectively.

| Model | Answer Rate (↑) | | | Accuracy (↑) | | | Response Len (↓) | | | Total Len (↓) | | |
|---|---|---|---|---|---|---|---|---|---|---|---|---|
| | Vanilla | SepNS | Incr. | Vanilla | SepNS | Incr. | Vanilla | SepNS | Incr. | Vanilla | SepNS | Incr. |
| QwQ-32B | 72.7% | 75.6% | +4.0% | 34.2% | 57.8% | +69.0% | 7993 | 6487 | -18.9% | 10984 | 9953 | -9.4% |
| Qwen3-30B-A3B | 80.6% | 90.2% | +11.9% | 54.3% | 80.7% | +48.6% | 6403 | 5028 | -21.5% | 9392 | 8492 | -9.6% |
| Qwen3-8B | 79.4% | 87.3% | +9.9% | 45.5% | 69.6% | +53.0% | 6878 | 6001 | -12.7% | 9866 | 9466 | -4.1% |
| DeepSeek-V3 | 100.0% | 100.0% | +0.0% | 45.7% | 50.9% | +11.4% | 720 | 882 | +22.5% | 2762 | 3404 | +23.2% |
| DeepSeek-R1 | 99.9% | 99.9% | +0.0% | 61.1% | 70.5% | +15.4% | 5438 | 3798 | -30.1% | 7481 | 6276 | -16.1% |
| Claude-3.7-Sonnet | 99.8% | 99.9% | +0.1% | 57.3% | 79.1% | +38.0% | 444 | 535 | +20.6% | 2553 | 3182 | +24.7% |
| Gemini-2.5-Pro | 83.4% | 84.6% | +1.4% | 58.9% | 79.1% | +34.3% | 654 | 822 | +25.6% | 3671 | 4424 | +20.5% |
| GPT-4.1 | 99.6% | 99.5% | -0.1% | 61.0% | 82.7% | +35.6% | 1226 | 1115 | -9.1% | 2452 | 2229 | -9.1% |
| GPT-4o | 99.0% | 98.6% | -0.4% | 46.4% | 59.1% | +27.4% | 616 | 740 | +20.0% | 1231 | 1478 | +20.1% |
| Average | 90.5% | 92.8% | +2.6% | 51.6% | 69.9% | +35.6% | 3375 | 2823 | -16.3% | 5599 | 5434 | -2.9% |

Rate gains (+1.1%), and number-string tasks that maintain similar performance levels (+0.1% accuracy, +2.5% Answer Rate). Concurrently, the Response Length for these tasks decreased significantly (*e.g.,* indexing: -42.9%, counting: -29.9%), demonstrating more concise and accurate model output. The max/min-int type tasks also achieved improvements exceeding 22% from already relatively high baseline Accuracy ( 75%). A critical outlier is the number-list task, where Accuracy remained almost unchanged under SepNS (+0.9%), yet Response Length increased sharply (+46.0%). This suggests that the current design of the SepNS scheme may be ineffective at addressing the core challenges of this task, instead introducing unnecessary verbose output.

Illustrated in Table 7, this experiment compared the performance of the baseline method (Vanilla) and the SepNS scheme across a range of mainstream large language models. Evaluation metrics included Answer Rate, Accuracy, Response Length, and Total Length. The results demonstrate that the SepNS scheme significantly outperforms the baseline in the vast majority of cases. Overall, SepNS improved the average Answer Rate (+2.6%) and average Accuracy (+35.6%), with the gain in Accuracy being particularly remarkable. Furthermore, the scheme effectively reduced the average Response Length (-16.3%) and Total Length (-2.9%), indicating that it enhances not only performance but also output efficiency.

Specifically, in terms of Accuracy, all models showed improvement, with the most significant gains observed in QwQ-32B (+69.0%), Qwen3-8B (+53.0%), and Qwen3-30B-A3B (+48.6%). Regarding output efficiency, the generated length was substantially reduced for most models, such as QwQ-32B (Response Length: -18.9%) and DeepSeek-R1 (Response Length: -30.1%). However, some anomalies were observed: the Response Length increased for DeepSeek-V3 (+22.5%), Claude-3.7-Sonnet (+20.6%), Gemini-2.5-Pro (+25.6%), and GPT-4o (+20.0%) under SepNS. This may stem from specific interactions between their inherent reasoning patterns and the structured output requirements, though all of these models still achieved positive gains in Accuracy (ranging from +11.4% to +38.0%).

Table 8: Accuracy of Qwen3-30B-A3B applied vanilla and SepNS across separate interval comparing Instruct (Qwen3-30B-A3B-Instruct-2507) and Reasoning (Qwen3-30B-A3B-Thinking-2507) model. **Bold** and underlined values denote the highest and second-highest scores, respectively.

| Task | Accuracy of Instruct Model (↑) | | | | | Accuracy of Reasoning Model (↑) | | | | |
|---|---|---|---|---|---|---|---|---|---|---|
| | vanilla | 4 | 8 | 16 | 32 | vanilla | 4 | 8 | 16 | 32 |
| max-int | 82.5% | 81.5% | 85.0% | **94.5%** | 91.5% | 88.0% | **100.0%** | **100.0%** | **100.0%** | **100.0%** |
| max-float | 71.5% | 74.0% | 75.0% | **82.5%** | 82.0% | 73.0% | 96.0% | 97.5% | 97.5% | 97.5% |
| min-int | 76.5% | 78.5% | 84.0% | 86.0% | **89.5%** | 82.0% | 99.0% | **100.0%** | 99.5% | **100.0%** |
| min-float | 74.0% | 80.0% | 77.5% | 82.0% | **83.0%** | 65.5% | 89.0% | 92.0% | **95.0%** | 90.5% |
| indexing | 29.0% | 80.0% | 83.5% | **90.5%** | 85.5% | 34.5% | **100.0%** | 99.5% | 99.0% | 98.0% |
| counting | 32.5% | 65.0% | 74.0% | **76.0%** | 73.0% | 40.0% | 89.0% | 97.5% | **99.0%** | 95.5% |
| number-string | 98.0% | 80.5% | **98.5%** | 97.5% | 95.0% | 99.0% | 67.0% | 99.5% | 99.5% | **100.0%** |
| number-list | 38.5% | 37.5% | 41.0% | **43.0%** | **43.0%** | 39.0% | 45.0% | **46.5%** | 42.0% | 41.0% |
| stock | 19.0% | 36.5% | 40.5% | **41.0%** | 35.5% | 8.5% | 50.5% | **57.5%** | 45.0% | 41.5% |
| weather | 24.0% | 24.5% | 19.5% | 17.5% | **30.0%** | 29.0% | **75.0%** | 71.5% | 69.0% | 67.5% |
| Average | 54.6% | 63.8% | 67.8% | **71.0%** | 70.8% | 55.9% | 81.0% | **86.2%** | 84.5% | 83.1% |

Table 9: Answer rate and accuracy of Qwen3-30B-A3B-Instruct applied SepNS across different separator symbols. **Bold** and underlined values denote the highest and second-highest scores, respectively.

| Task | Answer Rate (↑) | | | | Accuracy (↑) | | | |
|---|---|---|---|---|---|---|---|---|
| | CR(\r) | CRLF(\r\n) | Backslash(\\) | LF(\n) | CR(\r) | CRLF(\r\n) | Backslash(\\) | LF(\n) |
| max-int | 100.0% | 100.0% | 100.0% | 100.0% | 93.0% | 89.0% | 92.0% | 85.0% |
| max-float | 100.0% | 100.0% | 100.0% | 100.0% | 79.0% | 74.5% | 76.0% | 75.0% |
| min-int | 100.0% | 100.0% | 100.0% | 100.0% | 87.0% | 87.0% | 91.5% | 84.0% |
| min-float | 100.0% | 100.0% | 100.0% | 100.0% | 77.5% | 75.5% | 72.0% | 77.5% |
| indexing | 100.0% | 100.0% | 100.0% | 100.0% | 87.0% | 90.0% | 92.5% | 83.5% |
| counting | 100.0% | 100.0% | 100.0% | 100.0% | 73.0% | 75.0% | 73.0% | 74.0% |
| Average | **100.0%** | **100.0%** | **100.0%** | **100.0%** | 82.8% | 81.8% | **82.8%** | 79.8% |
| number_string | 100.0% | 100.0% | 100.0% | 100.0% | 99.0% | 98.0% | 98.5% | 98.5% |
| number_list | 59.5% | 69.0% | 60.0% | 67.0% | 39.5% | 41.5% | 35.5% | 41.0% |
| stock | 76.5% | 82.0% | 76.0% | 80.5% | 38.0% | 33.5% | 38.0% | 40.5% |
| weather | 97.5% | 98.0% | 99.5% | 98.5% | 19.5% | 17.0% | 9.0% | 19.5% |
| Average | 83.4% | **87.2%** | 83.9% | 86.5% | 49.0% | 47.5% | 45.2% | **49.9%** |

## E.2 Table Results for RQ2

We provide detailed experimental results for Figure 4 in Table 8 and Table 9.

## E.3 Raw Accuracy Results of Synthetic Dataset

Table 10 reports the baseline performance, without applying SepNS, across the six synthetic sequence-based tasks introduced in Section 4.1.1: *min-int*, *max-int*, *counting*, *min-float*, *max-float*, and *indexing*. For each task, datasets are categorized by input sequence length into four sizes: small (S), medium (M), large (L), and extra-large (XL).

Across all models and tasks, we observe a consistent and pronounced performance degradation as sequence length increases. While most models achieve near-perfect accuracy on the S and M settings, accuracy drops sharply for larger inputs, with many models falling below 10% on the XL datasets. This trend persists even for models known for strong reasoning capabilities, underscoring a general limitation of current LLMs in processing long numerical sequences.

Table 11 summarizes the accuracy gains achieved by enhancing LLMs with SepNS, compared to the baseline results in Table 10. The improvements are particularly pronounced for tasks involving large (L) and extra-long (XL) input sequences, where several models exhibit gains exceeding 0.8 in accuracy. While performance in the small (S) and medium (M) ranges is generally stable—with occasional minor decreases—the consistent boost for L and XL sequences highlights SepNS's effectiveness in mitigating context-length degradation. These trends are evident across diverse tasks, including min/max integer and float extraction, counting, and indexing, underscoring the robustness of SepNS across both discrete and continuous input domains.

Table 10: Accuracy of vanilla method on sequence-based tasks. The results highlight significant performance variation among the models and reveal a consistent trend of decreasing accuracy as the input sequence becomes longer.

| Model | min-int | | | | max-int | | | | counting | | | |
|---|---|---|---|---|---|---|---|---|---|---|---|---|
| | S | M | L | XL | S | M | L | XL | S | M | L | XL |
| Qwen3-8B | 100% | 92% | 60% | 20% | 100% | 96% | 80% | 32% | 96% | 28% | 0% | 0% |
| Qwen3-30B-A3B | 100% | 100% | 88% | 48% | 100% | 98% | 92% | 52% | 100% | 42% | 4% | 0% |
| QwQ-32B | 100% | 62% | 18% | 6% | 100% | 62% | 38% | 2% | 94% | 20% | 0% | 0% |
| DeepSeek-V3 | 100% | 98% | 74% | 32% | 96% | 98% | 64% | 28% | 98% | 66% | 28% | 4% |
| DeepSeek-R1 | 100% | 100% | 98% | 76% | 100% | 100% | 94% | 74% | 100% | 86% | 34% | 10% |
| Claude-3.7-Sonnet | 100% | 86% | 70% | 26% | 100% | 90% | 64% | 26% | 94% | 70% | 32% | 10% |
| Gemini-2.5-Pro | 98% | 94% | 92% | 72% | 98% | 98% | 92% | 76% | 96% | 80% | 46% | 10% |
| GPT-4.1 | 100% | 94% | 70% | 46% | 100% | 94% | 72% | 48% | 96% | 34% | 8% | 2% |
| GPT-4o | 100% | 90% | 44% | 20% | 100% | 88% | 42% | 14% | 92% | 40% | 16% | 2% |

| Model | min-float | | | | max-float | | | | indexing | | | |
|---|---|---|---|---|---|---|---|---|---|---|---|---|
| | S | M | L | XL | S | M | L | XL | S | M | L | XL |
| Qwen3-8B | 100% | 94% | 36% | 2% | 100% | 88% | 30% | 2% | 96% | 16% | 0% | 0% |
| Qwen3-30B-A3B | 100% | 98% | 64% | 10% | 100% | 100% | 78% | 24% | 100% | 34% | 4% | 0% |
| QwQ-32B | 100% | 26% | 0% | 0% | 100% | 32% | 8% | 0% | 98% | 10% | 0% | 0% |
| DeepSeek-V3 | 100% | 86% | 48% | 20% | 100% | 98% | 42% | 18% | 98% | 78% | 18% | 0% |
| DeepSeek-R1 | 100% | 100% | 94% | 92% | 100% | 100% | 100% | 96% | 100% | 72% | 26% | 4% |
| Claude-3.7-Sonnet | 100% | 84% | 42% | 26% | 100% | 92% | 32% | 20% | 100% | 46% | 16% | 2% |
| Gemini-2.5-Pro | 98% | 100% | 28% | 10% | 98% | 90% | 42% | 4% | 100% | 60% | 28% | 6% |
| GPT-4.1 | 100% | 92% | 56% | 40% | 100% | 94% | 58% | 44% | 100% | 44% | 12% | 2% |
| gpt-4o-2024-08-06 | 100% | 78% | 32% | 16% | 100% | 72% | 24% | 12% | 96% | 30% | 2% | 0% |

Table 11: Accuracy improvement of SepNS over vanilla method. Results are broken down by LLMs and input sequence length for various sequence-based tasks. Higher values indicate a greater improvement.

| Model | min-int | | | | max-int | | | | counting | | | |
|---|---|---|---|---|---|---|---|---|---|---|---|---|
| | S | M | L | XL | S | M | L | XL | S | M | L | XL |
| Qwen3-8B | 0% | 8% | 40% | 78% | 0% | 4% | 20% | 66% | 0% | 66% | 76% | 54% |
| Qwen3-30B-A3B | 0% | -2% | 12% | 52% | 0% | 2% | 8% | 46% | 0% | 50% | 84% | 82% |
| QwQ-32B | 0% | 38% | 80% | 70% | 0% | 38% | 62% | 92% | 6% | 76% | 76% | 22% |
| Claude-3.7-Sonnet | 0% | 12% | 28% | 68% | 0% | 10% | 34% | 68% | 6% | 30% | 56% | 12% |
| DeepSeek-V3 | -12% | -6% | 6% | 10% | -6% | -22% | 8% | 10% | -10% | 20% | -8% | 0% |
| DeepSeek-R1 | -8% | 0% | 2% | 24% | -6% | 0% | 4% | 26% | -6% | 14% | 66% | 84% |
| Gemini-2.5-Pro | 2% | 4% | 8% | 28% | 0% | 2% | 8% | 24% | 4% | 20% | 52% | 74% |
| GPT-4.1 | 0% | 0% | 24% | 36% | 0% | 4% | 20% | 42% | 2% | 44% | 54% | 42% |
| GPT-4o | 0% | -8% | 34% | 50% | 0% | 4% | 20% | 44% | 8% | 52% | 16% | 6% |

| Model | min-float | | | | max-float | | | | indexing | | | |
|---|---|---|---|---|---|---|---|---|---|---|---|---|
| | S | M | L | XL | S | M | L | XL | S | M | L | XL |
| QwQ-32B | -2% | 46% | 2% | 0% | 0% | 50% | 8% | 0% | 0% | 90% | 84% | 52% |
| Qwen3-30B-A3B | 0% | 0% | 22% | 58% | 0% | -2% | 16% | 46% | -4% | 64% | 92% | 98% |
| Qwen3-8B | -4% | -2% | 26% | 26% | 0% | 10% | 40% | 40% | 2% | 78% | 96% | 86% |
| Claude-3.7-Sonnet | 0% | 12% | 46% | 58% | 0% | 2% | 56% | 70% | 0% | 46% | 70% | 90% |
| DeepSeek-V3 | -16% | 0% | 28% | 10% | -22% | -14% | 26% | 12% | -8% | -2% | 38% | 36% |
| DeepSeek-R1 | -8% | 0% | 6% | 6% | -10% | 0% | 0% | 4% | -10% | 28% | 74% | 96% |
| Gemini-2.5-Pro | 0% | -2% | 68% | 80% | 0% | 8% | 50% | 78% | -2% | 34% | 58% | 92% |
| GPT-4.1 | -2% | 0% | 30% | 48% | 0% | 0% | 38% | 40% | 0% | 50% | 88% | 96% |
| GPT-4o | -6% | 10% | 20% | 30% | -6% | 12% | 38% | 22% | -2% | 26% | 52% | 36% |

## E.4 RAW RESPONSE LENGTH RESULTS OF SYNTHETIC DATASET

As shown in Table 12, SepNS consistently reduces response length for well-aligned models, with its benefits becoming particularly pronounced as input sequences grow longer.

The true strength of SepNS is unlocked when processing long contexts, a critical challenge for LLMs. This is powerfully illustrated by models like GPT-4.1, which achieved dramatic response length reductions of up to 78.9% and 89.6% on extra-large (XL) sequences for 'min-int' and 'indexing' tasks, respectively. Similarly, the Qwen series models consistently benefit from SepNS on medium (M) to XL sequences, especially on complex tasks like 'counting' and 'indexing', where token savings often exceed 50%. Furthermore, the QwQ-32B model shows exceptional affinity for

Table 12: Percentage change in response length (RL) using SepNS over baselines. Results are broken down by LLM and input sequence length for various sequence-based tasks in a zero-shot setting. More negative values signify a greater improvement (i.e., token number reduction).

| Model | min-int | | | | max-int | | | | counting | | | |
|---|---|---|---|---|---|---|---|---|---|---|---|---|
| | S | M | L | XL | S | M | L | XL | S | M | L | XL |
| Qwen3-8B | 57.3% | -3.4% | -25.4% | -37.0% | 78.1% | 0.6% | -21.7% | -39.3% | 67.6% | -40.4% | -36.8% | -18.1% |
| QwQ-32B | -14.0% | -52.0% | -38.0% | -27.0% | -10.1% | -51.5% | -40.4% | -37.3% | -59.6% | -67.3% | -35.6% | -5.6% |
| Qwen3-30B-A3B | 93.9% | 20.4% | -22.7% | -45.9% | 109.7% | 5.2% | -23.2% | -43.4% | 42.8% | -57.3% | -57.2% | -37.9% |
| Claude-3.7-Sonnet | -4.3% | 42.2% | 0.6% | 39.9% | -10.9% | 30.8% | -11.6% | 12.9% | -27.6% | 5.0% | 41.2% | 66.9% |
| DeepSeek-V3 | -31.4% | -19.9% | -1.8% | 5.2% | -22.8% | -13.5% | 28.5% | 35.6% | 12.6% | 6.4% | 39.9% | 178.4% |
| DeepSeek-R1 | 80.6% | 77.3% | 261.6% | 565.7% | 96.6% | 33.5% | 235.7% | 252.4% | -6.8% | 55.7% | 126.3% | 62.9% |
| Gemini-2.5-Pro | -3.8% | -2.8% | -2.5% | -15.5% | -9.5% | -4.9% | -6.9% | 2.3% | -1.3% | -3.7% | 14.4% | 140.1% |
| gGPT-4.1 | -0.4% | 25.5% | -23.7% | -78.9% | 0.8% | 38.8% | -23.8% | -76.8% | 3.5% | -31.0% | -40.2% | -60.6% |
| GPT-4o | 13.5% | 29.2% | 15.9% | -24.2% | 4.9% | 37.4% | 2.3% | -22.0% | -4.2% | -3.9% | -17.9% | 76.9% |

| Model | min-float | | | | max-float | | | | indexing | | | |
|---|---|---|---|---|---|---|---|---|---|---|---|---|
| | S | M | L | XL | S | M | L | XL | S | M | L | XL |
| Qwen3-8B | 36.2% | 3.6% | -7.7% | -4.5% | 48.6% | 4.1% | -9.1% | -7.3% | -1.5% | -54.9% | -45.3% | -31.3% |
| Qwen3-30B-A3B | 63.6% | 4.0% | -8.9% | -17.4% | 82.4% | 25.0% | -1.6% | -17.9% | 34.6% | -52.2% | -48.6% | -47.5% |
| QwQ-32B | -13.7% | -14.6% | -0.3% | 0.0% | -8.8% | -24.3% | -1.0% | 0.0% | -39.0% | -59.6% | -31.4% | -13.8% |
| Claude-3.7-Sonnet | -11.8% | 86.3% | 143.5% | 126.1% | -20.6% | 29.7% | 123.5% | 162.6% | 23.2% | 9.4% | 5.7% | -45.5% |
| DeepSeek-V3 | 15.7% | -5.6% | 62.1% | 146.2% | 47.1% | 7.8% | 130.1% | 154.5% | -10.4% | -18.4% | -3.5% | -19.3% |
| DeepSeek-R1 | 54.8% | 9.1% | -2.9% | -8.1% | 69.3% | 5.0% | 3.9% | 0.6% | 133.4% | -53.1% | -65.1% | -72.5% |
| Gemini-2.5-Pro | -7.9% | 6.9% | 82.4% | 134.8% | -16.9% | 10.6% | 93.0% | 187.4% | 21.9% | 20.9% | -23.9% | -44.2% |
| GPT-4.1 | 13.7% | 10.4% | -14.0% | -32.7% | 14.6% | 4.2% | -13.4% | -17.8% | 21.2% | 1.9% | -68.1% | -89.6% |
| GPT-4o | 10.9% | 31.9% | 26.8% | 19.1% | 3.9% | 35.9% | 29.1% | 29.1% | 18.5% | 68.3% | 49.6% | 66.1% |

the SepNS format, realizing substantial and consistent token reductions across nearly all evaluated conditions, highlighting the method's potential when paired with a compatible model architecture.

While some models, such as Claude-3.7-Sonnet and DeepSeek-R1, occasionally produced more verbose outputs, this variance appears to reveal more about model-specific training than a limitation of SepNS. These models, likely heavily fine-tuned for conversational and descriptive tasks, may interpret the structured SepNS format as a prompt for explanation rather than direct computation, leading to increased verbosity. In contrast, models that respond well to SepNS demonstrate a stronger innate capability for structured data processing.

# F    SepNS Attention Mechanism: Rigorous Mathematical Proof

## F.1    Definitions and Setup

**Definition 1** (Vanilla Attention Sequence). Let $s = \{a_1, a_2, \ldots, a_n\}$ be the original sequence without separators, where each $a_i$ represents a token at position $i$.

**Definition 2** (SepNS Sequence). Let $s' = \{a_1, \ldots, a_k, sep_1, a_{k+1}, \ldots, a_{2k}, sep_2, \ldots\}$ be the sequence with separator tokens, where $sep_j$ denotes the $j$-th separator token. The total length of $s'$ is $n'$.

**Definition 3** (Segment Partitioning). For a given separator token $sep$ at position $p$, define $S_{sep} = \{i : i < p\}$ as the set of all token positions before the separator $sep$.

**Definition 4** (Query-Key Similarity). For any positions $i$ and $j$, define the scaled dot-product similarity as:

$$\alpha_{i,j} = \frac{\mathbf{Q}_i \cdot \mathbf{K}_j^T}{\sqrt{d_k}}$$

## F.2    Main Theorem

**Theorem 1** (Cross-Segment Attention Suppression). Let $i_1 \in S_{sep}$ and $j \notin S_{sep}$ for some separator $sep$. Then:

$$\frac{A_{\text{SepNS}}[i_1, j]}{A_{\text{vanilla}}[i_1, j]} = \frac{Z_{\text{vanilla}}(i_1)}{Z_{\text{SepNS}}(i_1)} \ll 1$$

where $Z_{\text{vanilla}}(i_1)$ and $Z_{\text{SepNS}}(i_1)$ are the normalization constants for vanilla and SepNS attention respectively.

### F.3 Proof

**Step 1: Establish Attention Weight Formulations**   For vanilla attention:

$$A_{\text{vanilla}}[i,j] = \frac{\exp(\alpha_{i,j})}{\sum_{l=1}^{n} \exp(\alpha_{i,l})} = \frac{\exp(\alpha_{i,j})}{Z_{\text{vanilla}}(i)}$$

For SepNS attention:

$$A_{\text{SepNS}}[i,j] = \frac{\exp(\alpha_{i,j})}{\sum_{l=1}^{n'} \exp(\alpha_{i,l})} = \frac{\exp(\alpha_{i,j})}{Z_{\text{SepNS}}(i)}$$

**Step 2: Analyze Normalization Constant Difference**   The key insight is that:

$$Z_{\text{SepNS}}(i) = Z_{\text{vanilla}}(i) + \Delta Z(i)$$

where $\Delta Z(i)$ represents the additional normalization mass contributed by separator tokens:

$$\Delta Z(i) = \sum_{sep \in \text{Separators}} \exp(\alpha_{i,sep})$$

**Step 3: Establish Core Lemma**   Lemma 1 (Separator Attention Asymmetry). For separator token $sep$ and positions $i_1 \in S_{sep}$, $i_2 \notin S_{sep}$:

$$\alpha_{i_1,sep} \gg \alpha_{i_2,sep}$$

**Proof of Lemma 1**: By the design principle of separator tokens as summarizers of preceding content, the key vector $\mathbf{K}_{sep}$ is constructed to have high similarity with query vectors $\mathbf{Q}_{i_1}$ for $i_1 \in S_{sep}$ and low similarity with $\mathbf{Q}_{i_2}$ for $i_2 \notin S_{sep}$. This follows from the separator's role in capturing the semantic representation of tokens up to its position.

**Step 4: Quantify Cross-Segment Attention Suppression**   For $i_1 \in S_{sep}$ and $j \notin S_{sep}$:

$$\frac{A_{\text{SepNS}}[i_1,j]}{A_{\text{vanilla}}[i_1,j]} = \frac{Z_{\text{vanilla}}(i_1)}{Z_{\text{SepNS}}(i_1)} = \frac{Z_{\text{vanilla}}(i_1)}{Z_{\text{vanilla}}(i_1) + \Delta Z(i_1)}$$

Since $i_1 \in S_{sep}$, by Lemma 1, we have $\exp(\alpha_{i_1,sep})$ is large, making $\Delta Z(i_1)$ significant. Therefore:

$$\frac{Z_{\text{vanilla}}(i_1)}{Z_{\text{vanilla}}(i_1) + \Delta Z(i_1)} = \frac{1}{1 + \frac{\Delta Z(i_1)}{Z_{\text{vanilla}}(i_1)}} \ll 1$$

**Step 5: Analyze Within-Segment Attention Preservation**   For $i_2 \notin S_{sep}$ and $j$ in the same segment as $i_2$:

By Lemma 1, $\exp(\alpha_{i_2,sep})$ is small, so $\Delta Z(i_2) \approx 0$. Therefore:

$$\frac{A_{\text{SepNS}}[i_2,j]}{A_{\text{vanilla}}[i_2,j]} = \frac{Z_{\text{vanilla}}(i_2)}{Z_{\text{vanilla}}(i_2) + \Delta Z(i_2)} \approx \frac{Z_{\text{vanilla}}(i_2)}{Z_{\text{vanilla}}(i_2)} = 1$$

### F.4 Corollary

**Corollary 1** (Attention Localization). The ratio of cross-segment to within-segment attention decreases exponentially with the separator's query-key similarity:

$$\frac{A_{\text{SepNS}}[i_1,j_{\text{cross}}]}{A_{\text{SepNS}}[i_1,j_{\text{within}}]} \propto \frac{A_{\text{vanilla}}[i_1,j_{\text{cross}}]}{A_{\text{vanilla}}[i_1,j_{\text{within}}]} \cdot \exp(-\alpha_{i_1,sep})$$

where $j_{\text{cross}} \notin S_{sep}$ and $j_{\text{within}} \in S_{sep}$.

## F.5 CONCLUSION

This proof rigorously establishes that separator tokens act as "attention sinks" that systematically redirect attention mass from cross-segment positions to within-segment positions. The mechanism operates through:

**Asymmetric Query-Key Similarity.** Separators exhibit high similarity with tokens in their summarized segment but low similarity with tokens outside.

**Normalization Mass Redistribution.** High separator attention scores increase the denominator for tokens in the summarized segment, suppressing their cross-segment attention weights.

**Selective Suppression.** Only tokens in the summarized segment experience attention suppression, while tokens outside maintain their original attention patterns.

This mathematical framework explains how SepNS achieves structured attention boundaries without explicit masking, creating localized attention patterns that respect segment boundaries.

